# Metformin extends *C. elegans* lifespan through lysosomal pathway

Jie Chen[1,2†], Yuhui Ou[1†], Yi Li[1,2], Shumei Hu[1,2], Li-Wa Shao[1,2], Ying Liu[1]*

[1]State Key Laboratory of Membrane Biology, Institute of Molecular Medicine, Peking-Tsinghua Center for Life Sciences, Peking University, Beijing, China; [2]Academy for Advanced Interdisciplinary Studies, Peking University, Beijing, China

**Abstract** Metformin, a widely used first-line drug for treatment of type 2 diabetes (T2D), has been shown to extend lifespan and delay the onset of age-related diseases. However, its primary locus of action remains unclear. Using a pure in vitro reconstitution system, we demonstrate that metformin acts through the v-ATPase-Ragulator lysosomal pathway to coordinate mTORC1 and AMPK, two hubs governing metabolic programs. We further show in *Caenorhabditis elegans* that both v-ATPase-mediated TORC1 inhibition and v-ATPase-AXIN/LKB1-mediated AMPK activation contribute to the lifespan extension effect of metformin. Elucidating the molecular mechanism of metformin regulated healthspan extension will boost its therapeutic application in the treatment of human aging and age-related diseases.
DOI: https://doi.org/10.7554/eLife.31268.001

## Introduction

With the discovery that aging could be genetically regulated, numerous strategies have been employed to extend lifespan in model organisms, including pharmacologic and dietary interventions (*Longo et al., 2015*). Identification of a chemical or pharmacological manipulation that could target human aging and lower the risks associated with age-related diseases becomes a central goal of aging research. Administration of metformin, a first-line drug for treatment of type 2 diabetes (T2D), has been shown to extend lifespan in *C. elegans* and mice (*Anisimov et al., 2008*; *Cabreiro et al., 2013*; *De Haes et al., 2014*; *Martin-Montalvo et al., 2013*; *Onken and Driscoll, 2010*; *Wu et al., 2016*). In addition, metformin has been shown to ameliorate diabetic and cardiovascular diseases in patients (*Scarpello, 2003*). Metformin also lowered the incidence of several other age-related diseases, such as cancer, metabolic syndrome and cognitive disorders (*Foretz et al., 2014*). Due to its broad range of health benefits and little side effects, a clinical trial named TAME (Targeting Aging with Metformin) was proposed to evaluate metformin's protective effects against human aging and age-related diseases (*Barzilai et al., 2016*). However, despite its intriguing benefits to promote healthy aging, the underlying mode of action of metformin is not well understood and a subject of extensive debate.

Metformin is generally believed to act through activation of AMP-activated protein kinase (AMPK) (*Fryer et al., 2002*; *Hawley et al., 2002*; *Zhou et al., 2001*), a principal energy sensor that when activated, switches on catabolic pathways such as glycolysis and fatty acid oxidation to produce more ATP (*Burkewitz et al., 2014*; *Hardie et al., 2012*). It was proposed that metformin might act through inhibition of mitochondrial electron transport chain (ETC) Complex I (*El-Mir et al., 2000*; *Foretz et al., 2014*; *Owen et al., 2000*), resulting in a change of the AMP/ATP ratio and ultimately activating AMPK. However, this idea has been challenged recently (*He and Wondisford, 2015*), especially when physiological/low concentration (~70 uM) of metformin, which cannot induce AMP/ATP change, is still able to activate AMPK (*Cao et al., 2014*). An alternative model, in which metformin activates AMPK through the lysosome-dependent pathway was proposed (*Zhang et al., 2016*).

*For correspondence:
ying.liu@pku.edu.cn

†These authors contributed equally to this work

Competing interests: The authors declare that no competing interests exist.

**eLife digest** As humans are living for longer, age-related diseases – including cancer, diabetes, cardiovascular diseases and cognitive disorders – are becoming more common. Many research groups are therefore trying to find drugs that might prevent these diseases or make them less harmful.

A drug called metformin has been shown to extend the healthy lifespan of animals such as mice and the roundworm *Caenorhabditis elegans*. The drug is also currently used to treat type 2 diabetes in humans and may help to prevent some other age-related diseases. However, it is still not clear exactly what effects metformin has on cells.

Healthy cells need to perform many 'metabolic' processes to produce the molecules necessary for survival. Cell compartments called lysosomes play a role in many of these processes because they digest unneeded biological molecules. Through a combination of biochemical and genetic experiments involving *C. elegans* and human cells, Chen, Ou et al. found that metformin coordinates two metabolic pathways that both depend on lysosomes. Metformin reduces the activity of a pathway (called mTOR) that boosts cell growth and the metabolic processes that build complex molecules. At the same time, the drug activates a metabolic pathway (called AMPK) that breaks down complex molecules. Overall, therefore, metformin organizes a switch from a more growth-promoting state to a more growth-restricting state.

Before metformin can be used more widely to treat human aging and age-related diseases, we need to understand how it works in even more detail. Further studies are required to discover which proteins metformin acts on inside cells, and a clinical trial has also been proposed to measure metformin's effects on healthy human aging and age-related diseases.

DOI: https://doi.org/10.7554/eLife.31268.002

In this model, metformin treatment induces lysosomal localization of the scaffold protein AXIN, which brings its associated protein liver kinase B1 (LKB1) to form a complex with v-ATPase-Ragulator on the surface of lysosome (*Zhang et al., 2016*; *Zhang et al., 2013*). LKB1 then phosphorylates the threonine residue in the activation loop of AMPK, leading to AMPK activation (*Hawley et al., 1996*; *Shaw et al., 2004*; *Woods et al., 2003*).

In addition to AMPK activation, metformin administration also inhibit mechanistic target of rapamycin complex 1 (mTORC1) (*Kalender et al., 2010*). mTORC1 constitutes another hub for energy and nutrient sensing, and switches on anabolism when activated (*Schmelzle and Hall, 2000*). The primary pathway for mTORC1 activation requires lysosome-localized Rag GTPases, which form RagA/B–RagC/D heterodimers and recruit mTORC1 to the surface of lysosome through directly binding to the Raptor subunit of mTORC1 (*Kim et al., 2008*; *Sancak et al., 2008*). v-ATPase-Ragulator complex on lysosomal surface is required for the spatial regulation and activation of mTORC1 by Rag GTPases (*Bar-Peled et al., 2012*; *Dibble and Manning, 2013*; *Jewell et al., 2013*; *Sancak et al., 2010*). Two distinct mechanisms have been proposed for metformin-induced mTORC1 inhibition: suppression of Rag GTPases (AMPK-independent mechanism) or AMPK-mediated phosphorylation of Raptor (regulatory associated protein of mTORC1) (AMPK-dependent mechanism) (*Howell et al., 2017*; *Kalender et al., 2010*).

Due to its short lifespan and ease of genetic manipulation, *C. elegans* becomes a powerful model organism to test the efficacy of metformin in promoting healthspan (*Burkewitz et al., 2014*). It has been shown that metformin treatment greatly extends worm lifespan and improves fitness, for example prolonging locomotory ability (*Onken and Driscoll, 2010*). However, several different mechanisms have been suggested for metformin's lifespan extension effect in *C. elegans*. For instance, metformin administration may mimic a dietary restriction (DR) metabolism (*Onken and Driscoll, 2010*). In addition, alteration of methionine metabolism in *C. elegans* has been shown to play a partial role (*Cabreiro et al., 2013*). Recently, metformin has been shown to inhibit mTORC1 due to restricted transit of RagA-RagC GTPase through nuclear pore complex (NPC) (*Wu et al., 2016*). It is possible that these observations reflect downstream consequences of a primary action of metformin. Therefore, understanding its direct mechanism of action worth further investigation.

Elucidating metformin's mode of action will significantly boost its application to target human aging and prevent age-related diseases.

## Results

### Metformin coordinates mTORC1 and AMPK through the lysosomal pathway

Intrigued by the discoveries that mTORC1 and AMPK share the common activator, v-ATPase-Ragulator complex (*Zhang et al., 2014*), and that metformin may directly act on the lysosomal pathway to promote AMPK activation (*Zhang et al., 2016*), we sought to set up a pure in vitro reconstitution system to explore the mechanism of metformin's action (*Figure 1*). We stably expressed LAMP1-RFP-FLAG in HEK293T cells to label lysosomes (*Figure 1A*) and performed FLAG pull-down to enrich lysosomes. The purity of lysosome preparation was confirmed by immunoblotting to test for the absence of early endosome, mitochondria, ER or nuclei (*Figure 1B*). Immuno-purified Raptor was then provided to the lysosomes. Upon amino acids stimulation, significant portion of Raptor accumulated on the lysosome (*Figure 1C,F*), indicating the activation of mTORC1 pathway. We then supplemented Concanavalin A (Con A), a v-ATPase inhibitor, or Metformin (Met) to the reconstitution system. Addition of Con A or metformin dissociated Raptor from the lysosome (*Figure 1D,G*). It has been proposed that AMPK might directly phosphorylates Raptor and thus inhibits mTORC1 (*Gwinn et al., 2008*). To test if metformin's effect on mTORC1 inhibition requires AMPK or not, we repeated the in vitro experiments in AMPK knockout MEF cells, and still observed the dissociation of Raptor (*Figure 1—figure supplement 1*). Taken together, these results suggest that metformin inhibits mTORC1 through the lysosomal pathway independent of AMPK, possibly mimicking Con A to inhibit v-ATPase. Lastly, we provided purified AXIN and LKB1 to the in vitro system. Immuno-purified LKB1 forms a complex with endogenous STRAD and MO25, two proteins essential for optimal LKB1 activity (*Figure 1—figure supplement 2*). Consistent with previous findings, AXIN/LKB1 were recruited to metformin- or Con A-treated lysosomes, and were sufficient to phosphorylate endogenous AMPK in an amino acid-independent manner (*Figure 1E,H*) (*Zhang et al., 2016*; *Zhang et al., 2013*).

It has been proposed that metformin might change the AMP/ATP ratio through inhibition of mitochondrial ETC complex I, ultimately resulting in AMPK activation (*El-Mir et al., 2000*; *Foretz et al., 2014*; *Owen et al., 2000*). To test if metformin could inhibit mitochondrial function and therefore activate mitochondrial stress response, we treated *C. elegans hsp-6p::gfp* reporter strain with metformin, or rotenone, a well-known mitochondrial ETC complex I inhibitor. Unlike rotenone, metformin treatment showed no reporter activation (*Figure 1—figure supplement 3A,B*). It should be noted that others have reported that metformin inhibits ETC complex I through a mechanism distinct from that of rotenone: metformin treatment increases ROS production, whereas rotenone decreases it (*De Haes et al., 2014*). Because our in vitro reconstitution system excludes mitochondrial contaminants, it highly suggests that metformin inhibits mTORC1 and activates AMPK mainly through the lysosomal pathway. Lastly, we directly measured the metformin's effect on lysosomal function with the use of a cathepsin assay. Cathepsin is a class of cysteine proteinase localized within lysosomes. A cathepsin assay is a fluorescence-based assay that cleaves cathepsin's substrate to release fluorescence. Consistent with our idea, metformin administration indeed impaired lysosomal function (*Figure 1—figure supplement 3C,D*).

### Metformin inhibits TORC1 pathway in *C. elegans*

Because our in vitro biochemical results suggested that metformin might act on the v-ATPase-Ragulator complex of the lysosomal pathway to coordinate mTORC1 inhibition and AMPK activation, we next tested if metformin promotes lifespan extension in *C. elegans* due to similar mechanisms. To confirm that metformin also inhibits TORC1 in *C. elegans*, we first examined phosphorylation status of RSKS-1, the human ribosomal protein S6 kinase B1 (S6K) homolog in *C. elegans*. We used an antibody that specifically detects Thr-389, a highly conserved phosphorylation site among species (*Figure 2—figure supplement 1A*). This antibody was validated with the used of *rsks-1* RNAi (*Figure 2—figure supplement 1B*). Notably, metformin treatment significantly decreased the level of RSKS-1 phosphorylation (*Figure 2A,B*), suggesting an inhibition of TORC1 pathway. We also

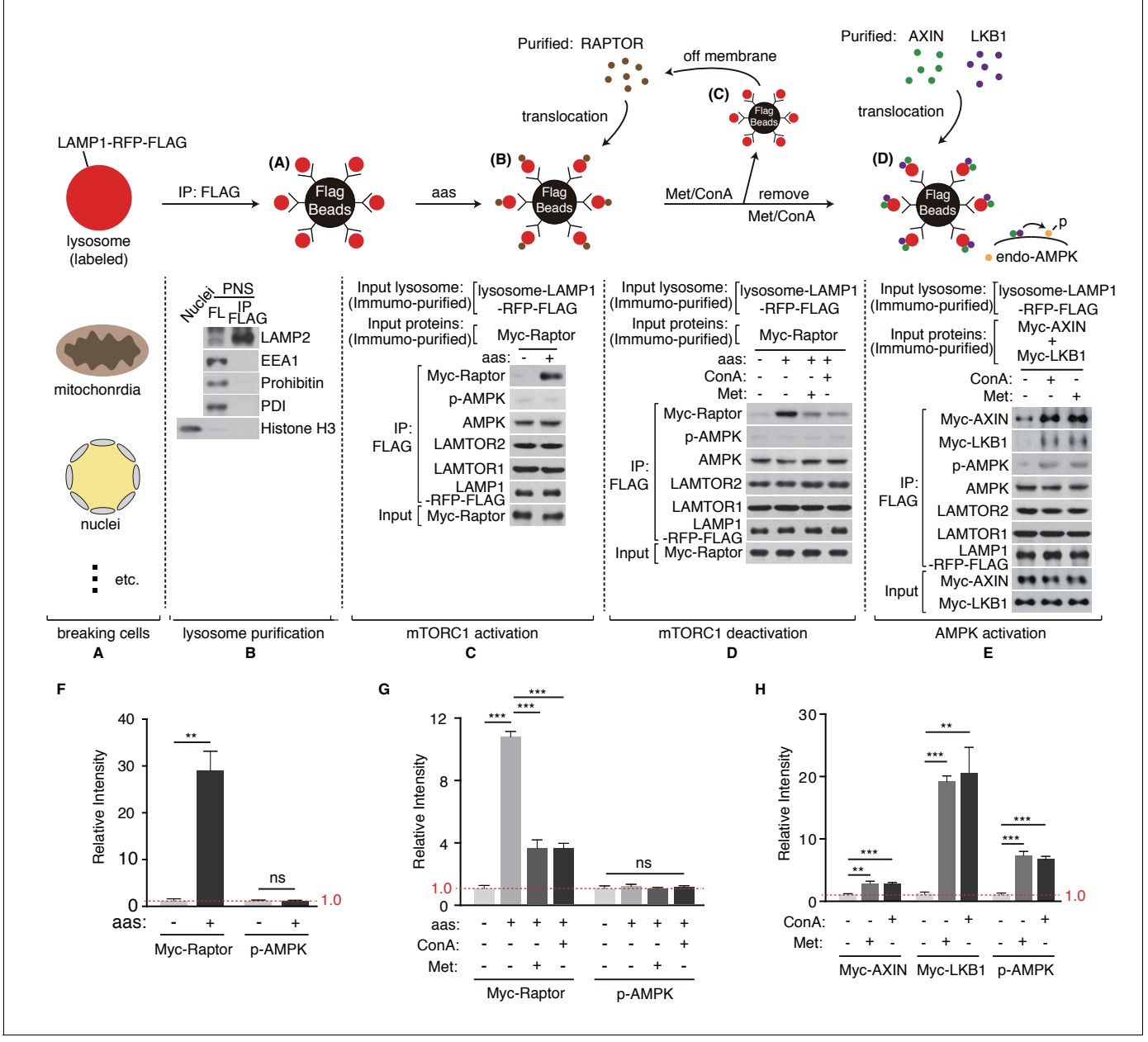

**Figure 1.** Metformin coordinates mTORC1 and AMPK on purified lysosome. (**A**) HEK293T cells stably expressing LAMP1-RFP-FLAG were mechanically broken. (**B**) Lysosomes were purified through immunoprecipitation (Organelle markers: LAMP2, lysosome; EEA1, early endosome; Prohibitin, mitochondria; PDI, ER; Histone H3, nucleus). (**C**) Lysosomal accumulation of purified Myc-raptor upon amino acids stimulation. (**D–E**) Lysosomal disassociation of Myc-raptor (**D**), lysosomal accumulation of Myc-AXIN/LKB1 and phosphorylation of AMPK (**E**) upon Concanamycin A (Con A) or Metformin (Met) treatment. (**F–H**) Quantifications of immunoblots in (**C–E**). Immunoblots of Myc-Raptor, Myc-AXIN and Myc-LKB1 were normalized to that of LAMP-RFP-FLAG, and immunoblots of p-AMPK were normalized to that of AMPK. Relative intensities of three independent biological replicates are shown as mean ± SEM. ns, no significant difference; *p<0.05; **p<0.01; ***p<0.001.

DOI: https://doi.org/10.7554/eLife.31268.003

The following figure supplements are available for figure 1:

**Figure supplement 1.** Metformin inhibits mTORC1 independent of AMPK.
DOI: https://doi.org/10.7554/eLife.31268.004

**Figure supplement 2.** Purified Myc-LKB1 forms a complex with endogenous STRAD and MO25.
DOI: https://doi.org/10.7554/eLife.31268.005

**Figure supplement 3.** Metformin targets lysosome in *C.elegans*.
DOI: https://doi.org/10.7554/eLife.31268.006

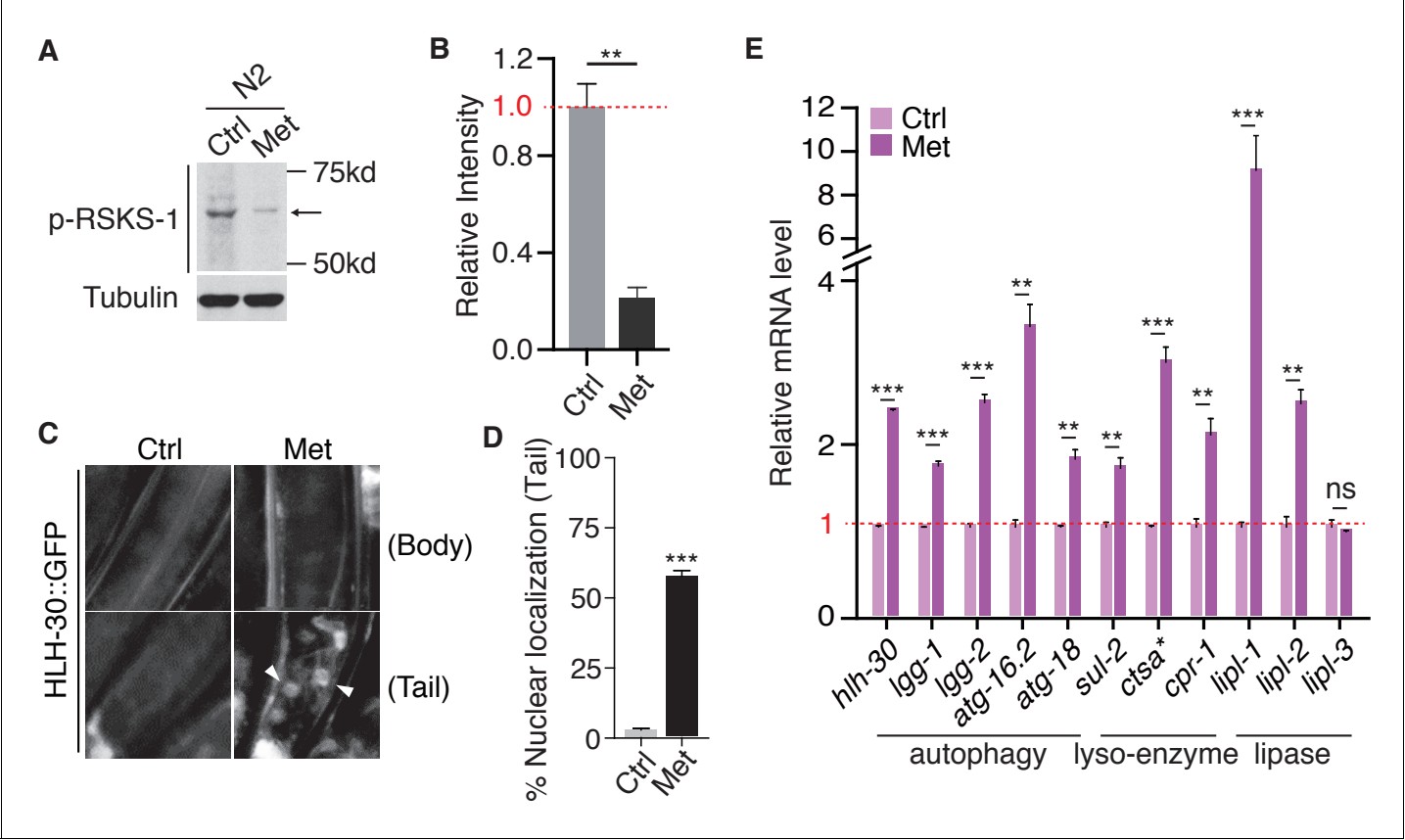

**Figure 2.** Metformin inhibits TORC1 pathway in *C.elegans*. (A) Representative western blotting of RSKS-1 phosphorylation in the presence or absence of Metformin. (B) Quantification of immunoblots in (A). Relative intensities of 3 independent biological replicates are shown as mean ± SEM. (C) Representative fluorescent images of HLH-30 nuclei localization in the presence or absence of metformin. (D) Percentage of worms with HLH-30 nuclear localization in (C) was quantified. Mean ± SEM of 3 independent biological replicates are shown (sample size:≥40 worms). (E) Q-PCR analysis of HLH-30 target genes in the presence or absence of metformin. ~300 worms were pooled in each sample. Data from three independent biological replicates are shown as mean ± SEM. ns, no significant difference; *p<0.05; **p<0.01; ***p<0.001.

DOI: https://doi.org/10.7554/eLife.31268.007

The following figure supplements are available for figure 2:

**Figure supplement 1.** RSKS-1, *C.elegans* homolog of S6K1 is phosphorylated at T389 residue.
DOI: https://doi.org/10.7554/eLife.31268.008
**Figure supplement 2.** Metformin inhibits TORC1 independent of AMPK in *C.elegans*.
DOI: https://doi.org/10.7554/eLife.31268.009

tested the subcellular localization of HLH-30, an ortholog of transcription factor EB (TFEB) that translocates to the nucleus upon TORC1 inhibition (*Lapierre et al., 2013*; *Perera and Zoncu, 2016*; *Settembre et al., 2011*). Indeed, HLH-30 translocated to the nucleus in the tail region of *C. elegans* upon metformin administration (*Figure 2C,D*). Once in the nucleus, HLH-30 regulates the transcription of genes related to autophagy, lysosome function and lipid hydrolysis (*Lapierre et al., 2013*; *O'Rourke and Ruvkun, 2013*). Consistently, we also observed up-regulation of transcript levels of *hlh-30* downstream targets upon metformin treatment (*Figure 2E*). Taken together, these results indicate that metformin inhibits TORC1 pathway in *C. elegans*.

We also tested if metformin's effect on TORC1 inhibition requires AMPK in *C. elegans*. Consistent with our results in AMPK-/- MEF cells (*Figure 1—figure supplement 1*), metformin treatment of *aak-2* (*Ce*.AMPK) loss-of-function mutants was still able to drive HLH-30 nuclear accumulation (*Figure 2—figure supplement 2A,B*) and elevate the expression of HLH-30 target genes (*Figure 2—figure supplement 2C*).

# Metformin extends healthspan partially through TORC1 inhibition

Inhibition of TORC1 has been shown to result in lifespan extension (*Jia et al., 2004*; *Vellai et al., 2003*; also reviewed by *Zoncu et al., 2011*). To distinguish if metformin induces lifespan extension solely through *Ce*.TOR inhibition, or the activation of *Ce*.AMPK also plays a role, we performed lifespan analysis on control worms or *daf-15* heterozygous mutants in the presence or absence of metformin (*Figure 3—figure supplement 1A*). Compared with the control worms, *daf-15* heterozygous mutants had a longer lifespan and a decreased level of RSKS-1 phosphorylation (*Figure 3—figure supplement 1B,C*), indicating that TORC1 inhibition extends *C. elegans* lifespan. More importantly, metformin treatment greatly extended the lifespan of *daf-15* heterozygous mutants (*Figure 3A*).

To investigate the metabolic effects of metformin on *daf-15* heterozygous mutants, we fed L4 worms with metformin and conducted Oil-Red-O (ORO) staining to detect neutral fat levels (*Figure 3—figure supplement 1D*). Similar to wild type animals, metformin administration further decreased neutral fat level in *daf-15* heterozygous mutants (*Figure 3B,C*). Aged worms start to show muscle deterioration and decrease locomotion rates (*Huang et al., 2004*; *Onken and Driscoll, 2010*). Metformin treatment significantly increased the locomotory ability (counted by the average bends of worm body per 60 s) and reduced age pigments of wild type worms and *daf-15* heterozygous mutants (*Figure 3—figure supplement 1D*, and *Figure 3D,E*), indicating that metformin also improves fitness of *daf-15* heterozygous mutants.

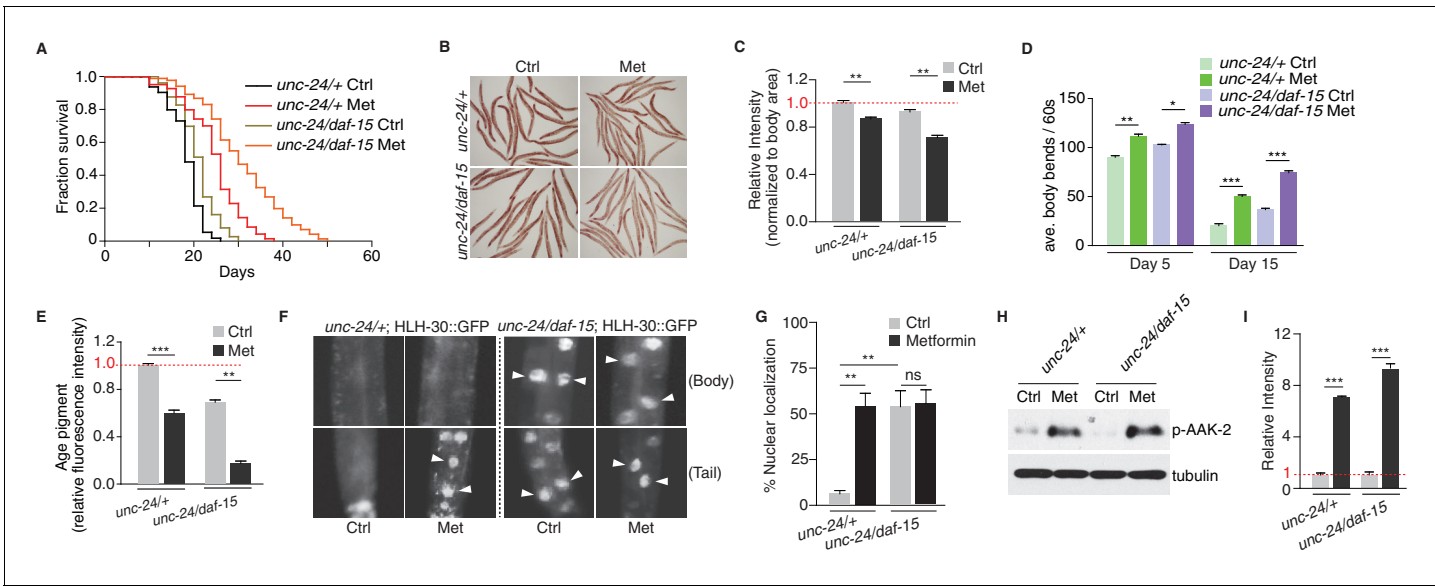

**Figure 3.** Metformin extends healthspan partially due to TORC1 inhibition. (A) Lifespan analysis of control worms or *daf-15* heterozygous mutants in the presence or absence of metformin. (B) Representative images of Oil-Red-O (ORO) staining of control worms or *daf-15* heterozygous mutants in the presence or absence of metformin. (C) Quantification of (B). Error bars represent mean ± SEM of 3 independent biological replicates (sample size: n ≥ 40 worms). (D–E) Locomotion (D) and age pigments (E) were measured in control worms or *daf-15* heterozygous mutants in the presence or absence of metformin. Mean ± SEM of 3 independent biological replicates are shown (sample size: n ≥ 20 worms for locomotion assay; n ≥ 40 worms for age pigments assay). (F) Representative fluorescent images of HLH-30 nuclei localization in control worms or *daf-15* heterozygous mutants in the presence or absence of metformin. Arrows indicate nuclear localized HLH-30::GFP. (G) Quantification of (F). Percentage of *unc-24/+*; HLH-30::GFP or *unc-24/daf-15*; HLH-30::GFP worms with nuclear accumulation of HLH-30 were counted. Error bars represent mean ± SEM of 3 independent biological replicates. (sample size: n ≥ 40 worms) (H) Representative western blotting of AAK-2 phosphorylation in the presence or absence of metformin. (I) Quantification of (H).~300 worms were pooled in each protein sample. Error bars represent mean ± SEM of 3 independent biological replicates. ns, no significant difference; *p<0.05, **p<0.01, ***p<0.001.

DOI: https://doi.org/10.7554/eLife.31268.010

The following figure supplements are available for figure 3:

**Figure supplement 1.** Deletion of *daf-15* suppresses TORC1 activity.
DOI: https://doi.org/10.7554/eLife.31268.011

**Figure supplement 2.** Metformin promotes longevity through a pathway additive to TORC1 inhibition.
DOI: https://doi.org/10.7554/eLife.31268.012

Metformin's beneficial effect on lifespan and fitness of *daf-15* heterozygous mutants could be due to a pathway additive to that of TORC1 inhibition (e.g. AMPK pathway), or simply due to a further suppression of TOR activity in *daf-15* heterozygous mutants. To test if metformin could further inhibit TORC1 activity in *daf-15* heterozygous mutants, we compared HLH-30 nuclear localization between control animals and *daf-15* heterozygous mutants in the presence or absence of metformin treatment. We found that in *daf-15* heterozygous mutants, significant portion of HLH-30 already localized in the nucleus, suggesting an inhibition of TORC1 activity. More importantly, metformin administration increased HLH-30 nuclear localization in control worms, but not in *daf-15* heterozygous mutants (*Figure 3F,G*). In addition, we performed lifespan experiments with loss-of-function mutants of known downstream targets of TORC1, which have been reported to mediate TORC1 inhibition-induced longevity (*Robida-Stubbs et al., 2012*). We found that metformin could still extend lifespans of *hlh-30*, *pha-4* or *skn-1* mutants (*Figure 3—figure supplement 2*). These results suggest that metformin may act on a pathway additive to that of TORC1 inhibition to confer fitness benefit. Given that metformin activated AMPK in *daf-15* heterozygous mutant to the same level as that in the control worms (*Figure 3H,I*), it is likely that metformin promotes longevity also through activation of AMPK.

## Metformin extends lifespan through lysosome-dependent activation of AMPK

v-ATPase-Ragulator complex may function as lysosomal acceptor for AXIN/LKB1, which translocate to lysosome and activate AMPK upon metformin treatment (*Figure 1E*). Thus, to test the lysosome-dependent AMPK activation for metformin-induced lifespan extension, we first performed lifespan analysis on *vha-3* (subunit of v-ATPase V0 domain), *vha-12* (subunit of v-ATPase V1 domain), *lmtr-3* (LAMTOR 3 subunit of Ragulator) and *lmtr-2* (LAMTOR 2 subunit of Ragulator) loss-of-function mutants (*Figure 4—figure supplement 1* and *Figure 4A*). VHA-3, VHA-12, LMTR-3 and LMTR-12 all localized on the lysosomes of *C. elegans*, as they have similar staining pattern with lysosomal marker LMP-1 (*Figure 4—figure supplement 2*). All these four mutants had impaired lysosomes, which could be rescued with the expression of corresponding gene (*Figure 4—figure supplement 3*). *vha-3* and *vha-12* mutants had longer lifespans compared with wild type animals, whereas lifespan of *lmtr-2* or *lmtr-3* mutants was comparable to that of the wild type animals (*Figure 4B and C* and *Figure 4—figure supplement 4A and B*). Lifespan extension in *vha-3* and *vha-12* mutants might be due to *Ce*.TOR inhibition, because v-ATPase is required for the spatial regulation and subsequent activation of TORC1. Consistently, failure of *lmtr-3* or *lmtr-2* to induce lifespan extension suggested that *Ce*.TOR pathway was not inhibited under *lmtr-3* or *lmtr-2* deficiency. Indeed, RNAi knockdown of *vha-3* but not *lmtr-3*, was sufficient to induce the nuclear accumulation of HLH-30 and activate HLH-30 downstream targets (*Figure 4—figure supplement 4C and D*, and *Figure 4D–F*). In addition, knocking down of *vha-3* but not *lmtr-3*, decreased the level of RSKS-1 phosphorylation (*Figure 4G and H*).

More importantly, metformin administration could not further induce the lifespan extension, nor AAK-2 activation in v-ATPase mutants *vha-3* and *vha-12*, or Ragulator mutants *lmtr-3* and *lmtr-2* (*Figure 4B–C and I–J* and *Figure 4—figure supplement 4A, B, E and F*), suggesting that metformin's effect on lifespan extension may also depend on the lysosomal pathway of AMPK activation. It should be noted that metformin even shortens the lifespan of *lmtr-3* or *lmtr-2* mutants (*Figure 4C* and *Figure 4—figure supplement 4B*). However, fractions of censored animals were not increased in metformin-treated *lmtr-3* or *lmtr-2* mutants (*Figure 4—figure supplement 4G*), suggesting that metformin did not make the mutants sick.

AXIN is a scaffold protein required for lysosomal localization of LKB1 and lysosome-dependent activation of AMPK (*Zhang et al., 2016*; *Zhang et al., 2013*). We could not successfully express AXL-1 (*C. elegans* ortholog of AXIN) with fluorescent tag in worms and test for its subcellular localization. Therefore, we expressed EGFP-tagged AXL-1 in mammalian cells and showed that AXL-1 only localized on the lysosome upon metformin treatment (*Figure 5A*). Consistently, without metformin administration, *axl-1* mutant alone did not perturb lysosomal function (*Figure 5B*). We then examined the role of AXIN in metformin-induced lifespan extension. Metformin treatment failed to activate AAK-2 (*Ce*.AMPK) in *axl-1* mutant (AXIN-deficient animals) (*Figure 5C,D*). Consistently, metformin was no longer able to extend lifespan of *axl-1* mutant animals (*Figure 5E*). Similar to *axl-1*, metformin could not extend lifespan of *par-4* (*C. elegans* ortholog of LKB-1) mutants as well

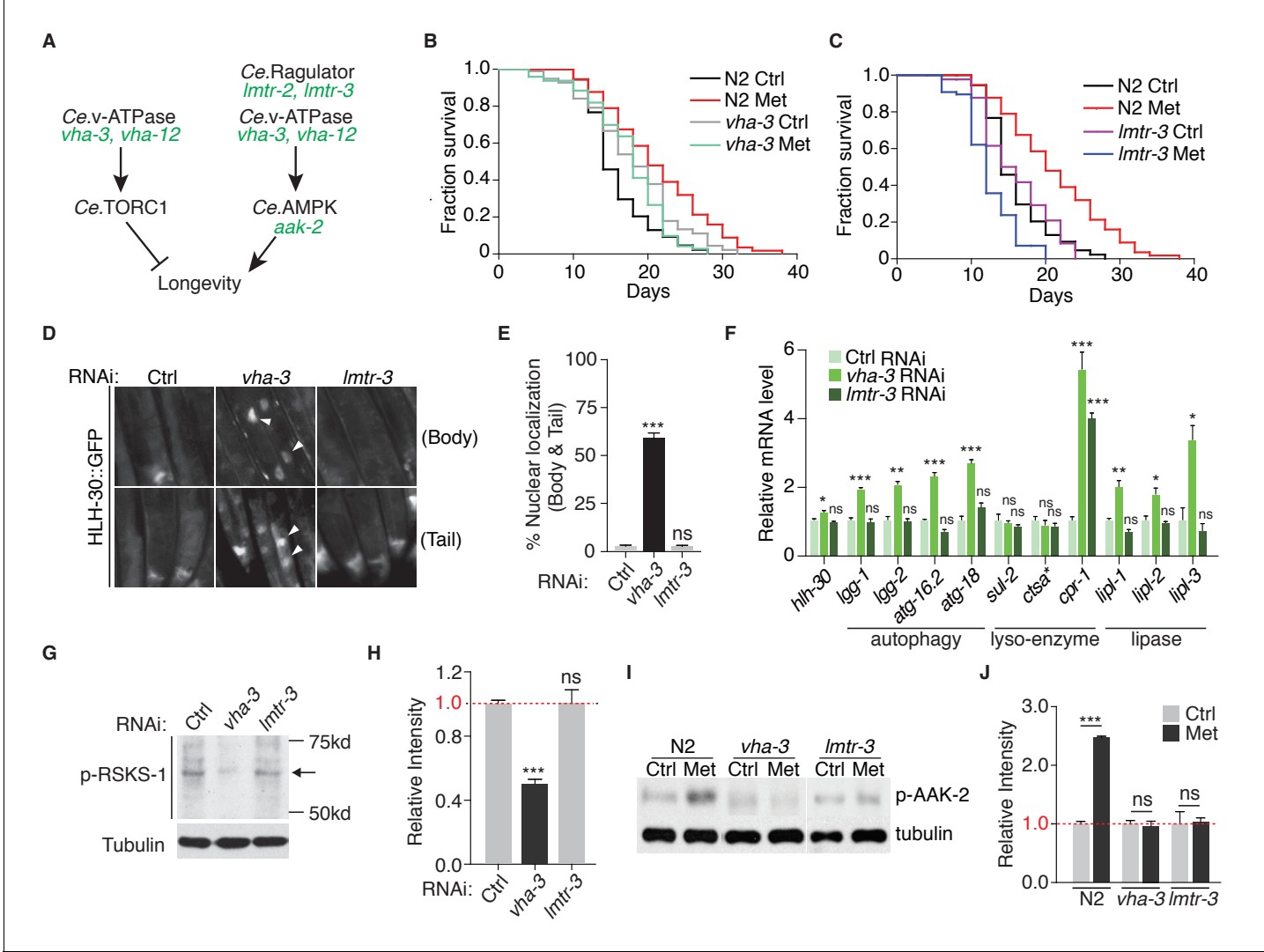

**Figure 4.** Metformin extends lifespan through v-ATPase-Ragulator-dependent activation of AMPK. (**A**) A scheme depicting genes within *C. elegans* TORC1 and AMPK pathways. (**B–C**) Lifespan analysis of wild type, *vha-3* (**B**), or *lmtr-3* (**C**) animals in the presence or absence of metformin. (**D**) Representative fluorescent images of HLH-30 nuclear localization in worms administrated with control, *vha-3* or *lmtr-3* RNAi. (**E**) Quantification of (**D**), percentage of worms with HLH-30 nuclear accumulation. Mean ± SEM of 3 independent biological replicates are shown (sample size: n ≥ 40 worms). (**F**) Q-PCR analysis of HLH-30 target genes in worms administrated with control, *vha-3* or *lmtr-3* RNAi. ~ 300 worms were collected for each mRNA sample. Data from 3 independent biological replicates are shown as mean ± SEM. (**G**) Representative immunoblots of RSKS-1 phosphorylation in worms administrated with control, *vha-3* or *lmtr-3* RNAi. (**H**) Quantification of (**G**). ~300 worms were pooled in each protein sample. Relative intensities of 3 independent biological replicates are shown as mean ± SEM. (**I**) Representative immunoblots of AAK-2 phosphorylation in wild type, *vha-3* or *lmtr-3* mutants, in the presence or absence of metformin. (**J**) Quantification of (**I**). ~300 worms were collected in each protein sample. Relative intensities of 3 independent biological replicates are shown as mean ± SEM. ns, no significant difference; *p<0.05; **p<0.01; ***p<0.001.
DOI: https://doi.org/10.7554/eLife.31268.013

The following figure supplements are available for figure 4:

**Figure supplement 1.** Genotyping of v-ATPase and Ragulator mutants.
DOI: https://doi.org/10.7554/eLife.31268.014

**Figure supplement 2.** Lysosomal localization of *C.elegans* v-ATPase-Ragulator proteins.
DOI: https://doi.org/10.7554/eLife.31268.015

**Figure supplement 3.** Mutation of v-ATPase or Ragulator perturbs lysosomal function in *C.elegans*.
DOI: https://doi.org/10.7554/eLife.31268.016

**Figure supplement 4.** Metformin extends lifespan through v-ATPase-Ragulator-dependent AMPK activation.
DOI: https://doi.org/10.7554/eLife.31268.017

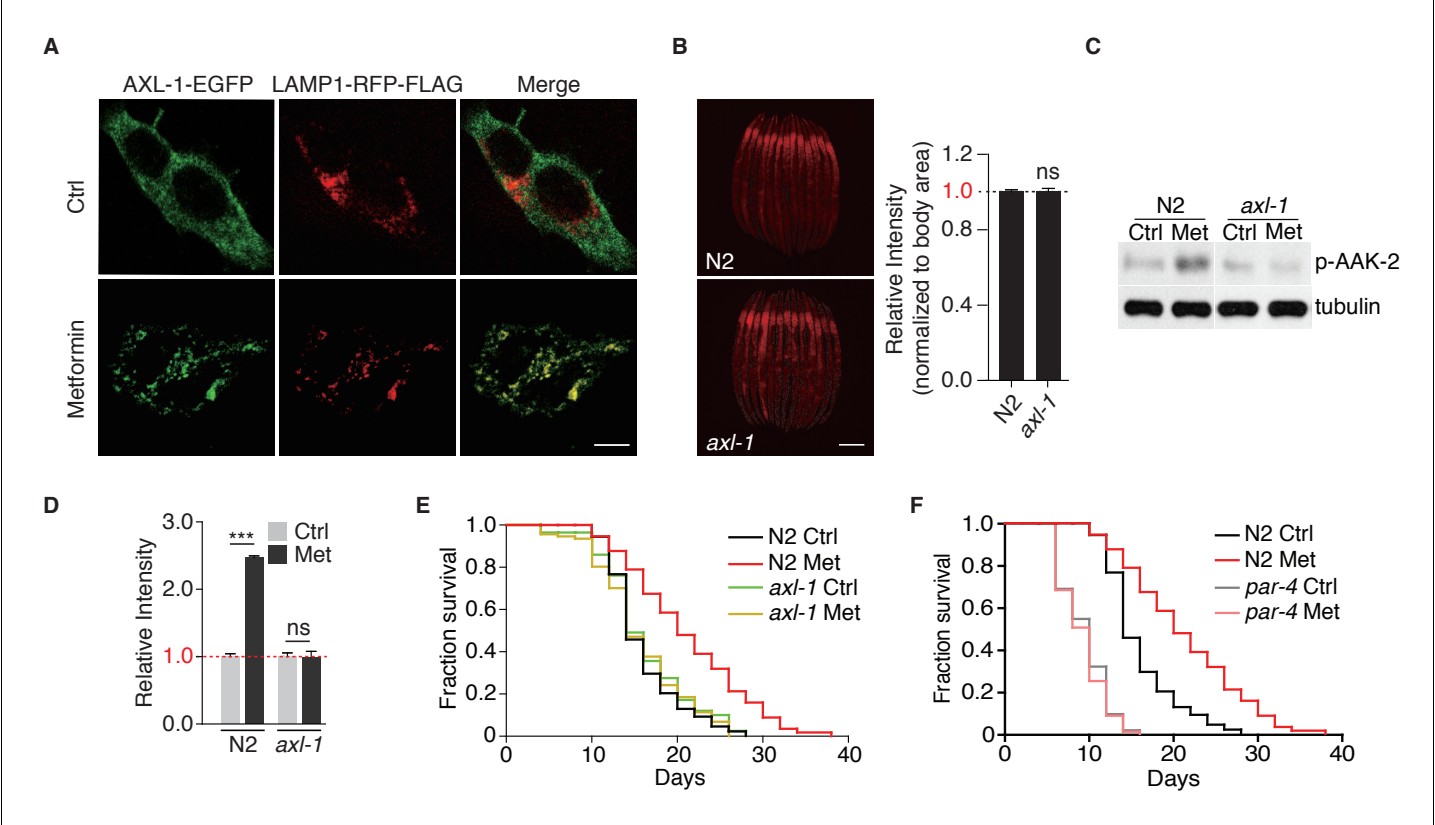

**Figure 5.** Metformin extends lifespan through *axl-1*-dependent activation of AMPK. (**A**) Representative fluorescent images to test co-localization of AXL-1 with LAMP1. HEK293T cells stably expressing AXL-1-EGFP and LAMP1-RFP-FLAG were cultured in the presence or absence of 2 mM metformin for 12 hr. Scale bar: 10 um. (**B**) Representative fluorescent images of Magic Red Cathepsin assay in wild type worms or *axl-1* mutants. Scale bar: 100 um. Error bars represent mean ± SEM of 3 independent biological replicates (sample size: n ≥ 40). (**C**) Representative immunoblots of AAK-2 phosphorylation in wild type or *axl-1* mutants with or without metformin treatment. (**D**) Quantification of (**C**).~300 worms were pooled in each protein sample. Relative intensities of 3 independent biological replicates are shown as mean ± SEM. (**E–F**) Lifespan analysis of wild type, *axl-1* (**E**), or *par-4* mutants (**F**) in the presence or absence of metformin. ns, no significant difference; *p<0.05; **p<0.01; ***p<0.001.
DOI: https://doi.org/10.7554/eLife.31268.018

(*Figure 5F*) (*Onken and Driscoll, 2010*). Taken together, these genetic evidences indicate that v-ATPase-Ragulator-AXIN-LKB1-based lysosome pathway is required for the metformin's effects of lifespan extension.

## Metformin attenuates age-related fitness decline through lysosome-dependent activation of AMPK

It has been shown that metformin triggers a dietary restriction-like state and extends *C. elegans* lifespan through AMPK pathway (*Onken and Driscoll, 2010*). Indeed, metformin administration induced a dietary restriction-like state to decrease neutral fat level in wild type animals (*Figure 6A, B*). Conversely, *vha-3* (v-ATPase V0), *lmtr-3* (Ragulator) and *axl-1* (AXIN) mutants retained the same levels of ORO staining in the presence or absence of metformin (*Figure 6A,B*), suggesting that the dietary restriction-like state triggered by metformin requires the lysosome-dependent activation of AMPK.

Metformin treatment also significantly increased the locomotory ability and lowered age pigments of wild type worms (*Figure 6C and G*), indicating that metformin promotes youthful physiology and fitness of *C. elegans*. Strikingly, mutation of *vha-3*, *lmtr-3* or *axl-1* impaired metformin's effects on locomotion and age pigments (*Figure 6D–G*), suggesting that lysosome-dependent activation of AMPK is also required for metformin-attenuated fitness decline in aged animals.

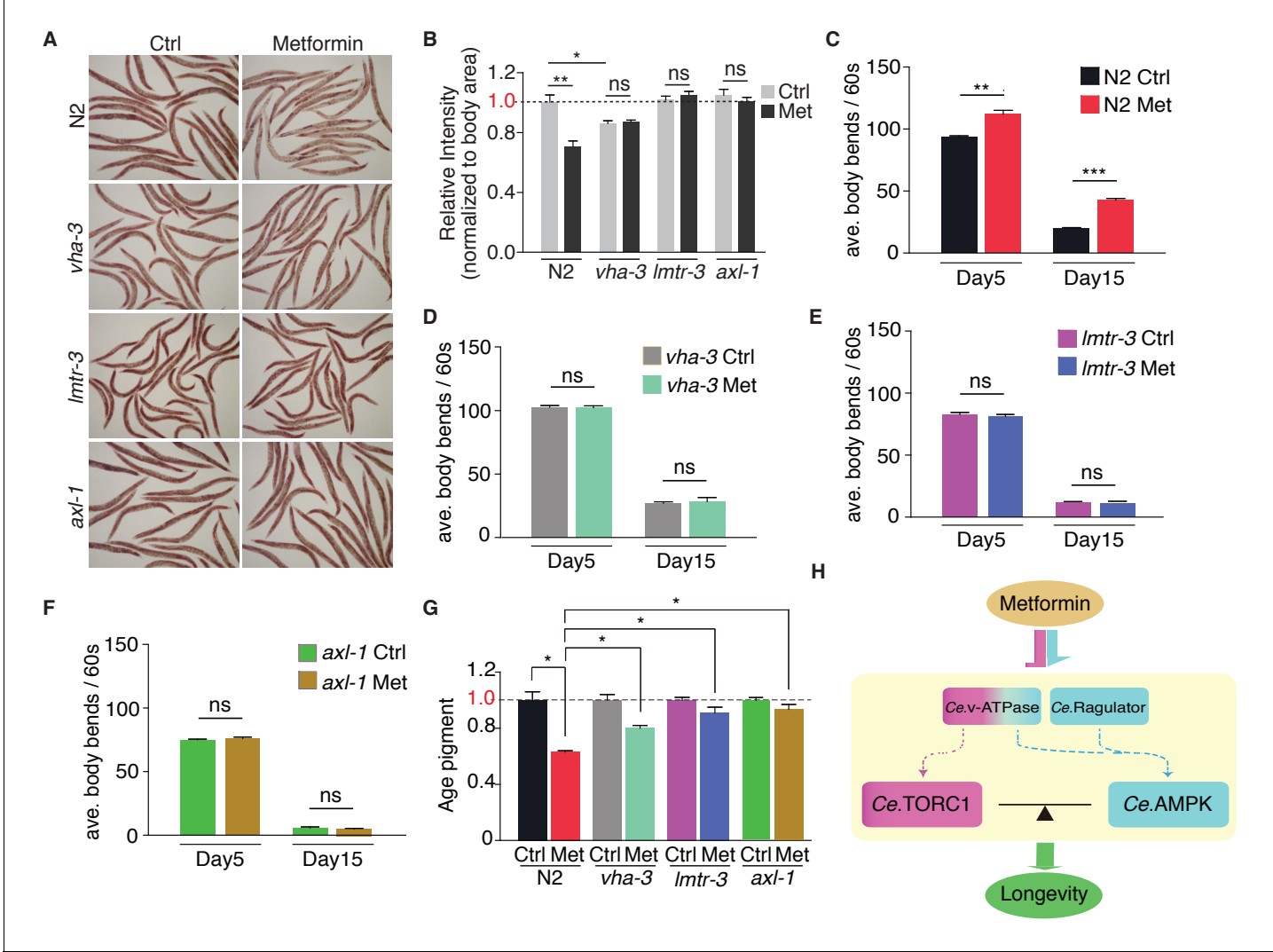

**Figure 6.** Metformin attenuates age-related fitness decline through lysosome-dependent activation of AMPK. Neutral fat deposition (**A–B**), locomotion (**C–F**) and age pigments (**G**) were measured in wild-type worms, or *vha-3*, *lmtr-3* or *axl-1* mutants in the presence or absence of metformin. Error bars represent mean ± SEM of 3 independent biological replicates. (sample size: n ≥ 20 worms for locomotion assay; n ≥ 40 worms for ORO staining or age pigments assay). (**H**) Metformin may target v-ATPase-Ragulator complex and promote longevity through coordination of *Ce*.TORC1 and *Ce*.AMPK.
DOI: https://doi.org/10.7554/eLife.31268.019

The following figure supplement is available for figure 6:

**Figure supplement 1.** Proposed model for metformin's action.
DOI: https://doi.org/10.7554/eLife.31268.020

## Discussion

Using a pure in vitro reconstitution system that excludes mitochondria, we showed that metformin coordinates mTORC1 inhibition and AMPK activation through lysosomal pathway. We further employed genetic manipulation to show that metformin extends *C. elegans* lifespan and attenuates age-related fitness decline via similar mechanism that requires v-ATPase-Ragulator-AXIN/LKB1 of the lysosomal pathway (*Figure 6H*). Metformin may function by targeting and priming v-ATPase-Ragulator complex on lysosome membrane, which serves as a hub to coordinate mTORC1 and AMPK pathways and govern metabolic programs. It is possible that metformin administration might result in a conformational change of v-ATPase-Ragulator complex, which dissociates mTORC1 from lysosome and allows the docking of AXIN/LKB1 for AMPK activation (*Figure 6—figure supplement*

*1*). It will be of particular interest in the future to test if v-ATPase is the direct target of metformin. Elucidating the molecular mechanism of metformin-mediated lifespan extension will boost its application in the treatment of human aging and age-related diseases.

## Materials and methods

### Cell culture

HEK293T cells used for protein purification, virus packaging and stable cell line generation, and AMPKα1/2 double knockout (DKO) MEF cells used for stable cell line generation were cultured with DMEM high glucose medium (HyClone, SH30243.01), supplemented with 10% FBS (Gibico, 10099–141) and 1% Penicillin/Streptomycin (HyClone SV30010). 293T HEK cells were obtained from ATCC, not authenticated, and mycoplasma negative. AMPKα1/2 double knockout (DKO) MEF cells were provided by Dr. Benoit Viollet, validated by immunoblotting and determined as mycoplasma negative.

### Generation of stable cell line

HEK293T cells and AMPKα1/2 DKO MEF cells stably expressing LAMP1-RFP-FLAG fusion protein, or HEK293T cells stably expressing AXL-EGFP and LAMP1-RFP-FLAG were generated with the lentiviral packaging and infection system. $0.8 \times 10^6$ HEK293T cells were seeded into one well of a 6-well-plate and cultured with 2 ml medium. After about 20 hr, 0.75 ug pMDLg/pRRE, 0.3 ug pRSV-Rev, 0.45 ug pCI-VSVG and 1.5 ug pLJM1-LAMP1-RFP-FLAG (addgene) plasmids were co-transfected into cells. Medium was changed 8 hr after transfection. Cells were then cultured for additional 24 hr to allow virus production, and 2 ml medium containing virus was harvested (virus packaging). 300 ul virus-containing medium was diluted with 900 ul fresh medium, and added into one well of a 12-well-plate containing HEK293T cells or AMPKα1/2 DKO MEF cells at ~40% confluence (virus infection). Several hours after infection, cells stably expressing LAMP1-RFP-FLAG were checked by RFP fluorescence. ~100% HEK293T cells or ~20% AMPKα1/2 DKO cells contain RFP signal. HEK293T LAMP1-RFP-FLAG stable cell line was then infected by virus carrying AXL-EGFP to generate double stable cell line.

### Protein purification

Recombinant proteins Myc-LKB1 (human), Myc-AXIN (mouse), and Myc-Raptor (human) were immunopurified from HEK293T cells. For purification of each protein, $20 \times 15$ cm dishes of HEK293T cells were cultured until 80% confluence. 28 ug of plasmid was transfected to each dish of cells with Lipofectamine 3000 Transfection Reagent. Medium was changed 8 hr after transfection. After additional 24 hr, medium was carefully discarded and cells were rinsed with PBS. 1.5 ml of TX-100 lysis buffer (20 mM Tris-HCl, 150 mM NaCl, 1 mM Na$_2$EDTA, 1 mM EGTA, 2.5 mM Sodium pyrophosphate, 1 mM β-glycerophosphate, 1% Triton X-100, PH = 7.4) supplemented with proteinase inhibitors was added to each dish of cells. Cells were incubated on ice for 10 min, and then scraped off the dishes. Cell lysate was sonicated at 60% Ampl for $5 \times 30$ s (SONICS VCX 130 PB, UK), and centrifuged at 4°C, 10000 rpm for 10 min. Supernatant was mixed with 500 ul Myc agarose beads, and rotated overnight at 4°C. Beads were washed three times with TX-100 lysis buffer and three times with fractionation buffer (50 mM KCl, 90 mM K-Gluconate, 1 mM EGTA, 5 mM MgCl$_2$, 50 mM Sucrose, 5 mM Glucose, 20 mM HEPES, pH = 7.4, 2.5 mM ATP and protease inhibitors were added right before usage), and resuspended in 800 ul of fractionation buffer, followed by addition of 10 ul 10 mg/ml Myc peptide. After rotating at 4°C for 4 hr, Myc tagged protein in 800 ul fractionation buffer was harvested, followed by addition of 160 ul glycerol, and stored at −20°C.

### Lysosome purification and in vitro reconstitution

LAMP1-RFP-FALG stable cells were cultured until confluence. For each sample, $1 \times 10$ cm dish of HEK293T cells or $5 \times 10$ cm dishes of MEF AMPKα1/2 DKO cells were scraped off the plate, and pelleted at 1000 rpm, room temperature for 3 min. Cell pellet was washed once with fractionation buffer, resuspended in 0.8 ml fractionation buffer, and mechanically broken by spraying six times through a 23G needle attached to a 1 ml syringe. Broken cells were spun down at 2000 g for 10 min at 4°C to pellet the nuclei, yielding a post-nuclei supernatant (PNS). The PNS was diluted to 2 ml

with fractionation buffer, and subjected to immunoprecipitation with 50 ul FLAG magnetic beads (60% slurry) for 2 hr at 4°C on a rotator. Lysosome on beads was washed three times and resuspended in 300 ul fractionation buffer, supplemented with 1x amino acids (combining 50x Gln-free MEM amino acids mixture with 100x Glutamine solution from Gibico), 250 uM GTP and 100 uM GDP. Lysosome was then rotated at 37°C, 650 rpm on a Thermomixer (Eppendorf, Germany) for 15 min (activation step for Raptor binding). 40 ul of purified Myc-Raptor protein was added, and the system was shaken for additional 25 min (lysosome binding step for Raptor). 10 uM Concanamycin A (ConA) or 300 uM Metformin (Met) was then provided into the reaction system, and incubated for 30 min (Raptor off lysosome membrane step). Lysosome coated magnetic beads were isolated from the reaction system, resuspended with 300 ul fractionation buffer supplemented with 80 ul purified Myc-AXIN and 20 ul purified Myc-LKB1 (formed a complex with endogenous STRAD and MO25), and then shaken on Thermomixer for 25 min (AXIN/LKB1 lysosome translocation and AMPK activation step).

## Western blotting

For in vitro reconstitution, each sample was prepared with 30 ul 2x laemmli sample buffer. Purified lysosomes were checked with LAMP2 (abcam ab25631), EEA1 (CST 3288), Prohibitin (Santa Cruz 28259), PDI (Santa Cruz 20132), and Histone H3 (Huaxingbio, HX1850) antibodies. Myc tagged proteins were detected with anti-Myc antibody (ImmunoWag YM3203). LAMP1-RFP-FLAG labeled lysosomes were probed with Flag antibody (Sigma F7425). T172 site phosphorylated AMPKα subunit and total AMPKα subunit were detected with p-AMPKα (CST 2535) and AMPKα (CST 2532) antibodies respectively. STRAD (Santa Cruz sc-34102) and MO25 (CST 2716 s) antibodies were used to detect Myc-LKB1-STRAD-MO25 complex.

For detection of AAK-2 or RSKS-1 phosphorylation, worms at indicated stage were lysed with 2x laemmli sample buffer equal to the volume of worm pellet. AAK-2 phosphorylation was probed with human p-AMPKα (T172) antibody (CST 2535). RSKS-1 phosphorylation was probed with human p-S6K1 (T389) antibody (CST 9205). Tubulin (abcam ab6161) was used as loading control.

## Nematode culture

Detailed information of *Caenorhabditis elegans* strains used in this paper was provided in *Supplementary file 1–2*. Worms were maintained on nematode growth medium (NGM) plate with OP50 as standard food. All worms, except for *pha-4* and *par-4* mutants, were cultured at 20°C. Metformin was added into molten agar at the concentration of 50 mM while preparing plates. Rotenone stock was prepared at 10 ug/ul in DMSO. 1 ul rotenone stock was diluted with 300ul M9 and added directly onto a 6 cm NGM plate and hood-dried.

RNA interference (RNAi) clones were grown at 37°C overnight in LB containing 50 ug/ml carbenicillin. 200 ul bacteria were then seeded onto NGM containing 1 mg/ml IPTG and 50 ug/ml carbenicillin, hood-dried and cultured overnight to induce dsRNA expression. L1 stage Worms were then seeded onto bacteria lawn, and raised until L4 stage for efficient knockdowns.

## Lifespan assays

~100 L4 stage (day 0 for lifespan assays) worms (except for *lmtr-2, lmtr-3, par-4* and *pha-*4) were transferred to plates with or without metformin. For *lmtr-2* or *lmtr-3* mutant strain,~250 worms were used, due to high percentage of censored worms. For *par-4* mutant, worms were cultured at 15°C until L4 stage (day 0 for lifespan assays), and ~100 worms were transferred to 25°C for lifespan assay. For *pha-4* mutant, worms were cultured at 25°C until day 1 adult (day 0 for lifespan assays), and ~100 worms were transferred to 15°C for lifespan assay. FUDR was used to prevent reproduction. Worms were transferred to new plates and counted every other day. FUDR was omitted after day 12. Animals that did not move when gently prodded were scored as dead. Animals that crawled off the plate or died from vulva bursting were censored.

## Oil-Red-O (ORO) staining

Worms were washed with 1x PBS and resuspended in 120 ul 1x PBS, followed by addition of 120 ul 2x MRWB buffer (160 mM KCl, 40 Mm NaCl, 14 mM Na$_4$EGTA, 1 mM Spermidine-HCl, 0.4 mM Spermine, 30 mM Na-PIPES, 0.2% β-mercaptoethanol, PH = 7.4) containing 2% PFA. Samples were

fixed by gently rocking at room temperature for 1 hr. Worms were then washed with 1x PBS to remove PFA, and resuspended in 300 ul 60% isopropanol to dehydrate for 15 min at room temperature. Worms were pelleted and resuspended with 1 ml 60% ORO solution, and rotated overnight at room temperature (Preparation of 60% ORO solution: commercial liquid ORO stain was diluted to 60% with $H_2O$, rocked overnight at room temperature, and filtrated with 0.22 um filter right before usage). Worms were settled and ORO staining solution was removed. 200 ul 1x PBS (with 0.01% triton X-100) was added to resuspend the worms. Worms were quickly transferred on to a glass slide and photographed with Zeiss Imager M2 microscope.

## Locomotion assay

Worms were transferred into a drop of M9 on a glass slide, and filmed with Zeiss Imager M2 microscope. Body bends were counted by reviewing each frame of the 60 s film.

## Age pigment fluorescence detection

Worms were mounted onto an agarose pad attached to a glass slide and photographed with Zeiss Imager M2 microscope. Fluorescence intensity was counted by Image J.

## Lysosome function assay

Magic Red stain (ImmunoChemistry Technologies #938, Bloomington, Minnesota) can be cleaved by cathepsin B, and generate red fluorescent substrate in functional lysosomes. Magic Red stain was prepared in 260x DMSO stock following the manufacturer's instructions. 3.8 ul stock was 5x diluted with M9, spread onto a well of 24-well-plate containing 1 ml NGM, and dried in hood. For metformin's effect on lysosomal function, wild type L1 stage worms were raised to L4 stage on control or metformin NGM plate containing Magic Red stain. For cathepsin assay, L4 stage worms were transferred onto Magic Red containing plate and cultured overnight. Worms were photographed with Zeiss Imager M2 microscope and quantified by Image J.

## Lysosomal localization assay

The longest CDS sequences of *lmp-1*, *vha-3/12* and *lmtr-2/3* were amplified from worm genomic DNA, and cloned into expression vector containing *rpl-28* promoter (whole body expression). LMP-1::GFP was initially integrated into *C. elegans* genome through UV irradiation, and worms were backcrossed for six times. mCherry-VHA-3/12 or mCherry-LMTR-2/3 was then injected into worms expressing LMP-1::GFP. Worms were photographed by Zeiss Imager M2 microscope. The longest CDS sequence of *axl-1* was amplified from worm genomic DNA and cloned into mammalian expression construct containing CMV promoter and C-terminal EGFP tag. HEK293T cells stably expressing AXL-1-EGFP and LAMP-1-RFP-FLAG were generated as described above. Cells were imaged by Zeiss LSM710 confocal microscope.

## Reporter assays

*hsp-6p::gfp* reporter worms were photographed by Zeiss Imager M2 microscope and the fluorescent intensity was quantified by Image J. *hlh-30p::hlh-30::gfp* reporter worms were photographed by Zeiss Imager M2 microscope and percentage of worms with nuclear localization of HLH-30 was counted.

To monitor HLH-30 nuclear localization in control or *daf-15* heterozygous mutant, we injected *hlh-30p::hlh-30::gfp* construct into control or *daf-15* heterozygous mutant respectively. Worms were photographed by Zeiss Imager M2 microscope. Percentage of worms with nuclear localization of HLH-30 was counted.

## RNA extraction and Q-PCR

Worms were washed off plates with M9 and resuspended with TRIzol Reagent and frozen by liquid nitrogen. Total RNA was isolated by chloroform extraction and isopropanol precipitation. 200 ng total RNA was used for reverse transcription with One-Step cDNA Synthesis Kit (TransGen Biotech, China). Real-time PCR was carried out using SYBR GREEN PCR Master Mix (Biorad, Hera Claes, California). Quantifications of transcripts were normalized to *rpl-32*.

## Q-PCR primers

| Target gene | Forward primer | Reverse primer |
| --- | --- | --- |
| rpl-32 | AGGGAATTGATAACCGTGTCCGCA | TGTAGGACTGCATGAGGAGCATGT |
| hlh-30 | CTCATCGGCCGGCGCTCATC | AGAACGCGATGCGTGGTGGG |
| lgg-1 | ACCCAGACCGTATTCCAGTG | ACGAAGTTGGATGCGTTTTC |
| lgg-2 | CTGCAAATTCCTAGTACCCGAG | CATAGAATTTGACACCATTGAGC |
| atg-16.2 | ATGTCATATCTGGATCTGCGG | ACGTTGCATCTGAAGAGCGTG |
| atg-18 | AAATGGACATCGGCTCTTTG | TGATAGCATCGAACCATCCA |
| sul-2 | ATGGCAGCAGAAGGCACCCG | GCCATTTTCCAACCATGCCAGTTGC |
| ctsa* | TTCTCCTCGAGGCGCGGGAT | TCCAACGCCAATTGGGGACTC |
| cpr-1 | CGCCAAGGACAAGCACTTCGGA | ACCTTGGCCTTTCCGGCGAC |
| lipl-1 | GTGACATTTGTTTTTCCATAT | AGCAAATTAAACCGACCAC |
| lipl-2 | AATACGAGTCAAATCATTGAA | GTAACACTCGTTTTTCCATAA |
| lipl-3 | ATGGGCAGGCAAATCCACCA | AGTTGTTCTGCGCAATTATA |

## Acknowledgements

We thank Dr. Sheng-Cai Lin for his generous gifts of plasmids and reagents. We thank Dr. Benoit Viollet for providing us AMPK knockout cells. Several *C. elegans* strains used in this work were provided by CGC, which is supported by the NIH-Officer of Research Infrastructure Programs, and the Japanese National BioResource Project. The work is supported by grants from the National Natural Science Foundation of China (grants 31422033 and 31471381) and the Ministry of Science and Technology of China (973 grants 2013CB910104), the Young Thousand Talents Program of China, and Peking-Tsinghua Center for Life Sciences awarded to YL.

## Additional information

### Funding

| Funder | Grant reference number | Author |
| --- | --- | --- |
| National Natural Science Foundation of China | 31422033 | Ying Liu |
| Ministry of Science and Technology of the People's Republic of China | 2013CB910104 | Ying Liu |
| National Natural Science Foundation of China | 31471381 | Ying Liu |
| Young Thousand Talents Plan of China | | Ying Liu |
| Peking-Tsinghua Center for Life Sciences | | Ying Liu |

The funders had no role in study design, data collection and interpretation, or the decision to submit the work for publication.

### Author contributions

Jie Chen, Conceptualization, Data curation, Formal analysis, Investigation, Visualization, Methodology, Writing—original draft, Project administration; Yuhui Ou, Data curation, Validation, Investigation; Yi Li, Validation, Investigation; Shumei Hu, Li-Wa Shao, Investigation; Ying Liu, Conceptualization, Supervision, Funding acquisition, Visualization, Methodology, Writing—original draft, Writing—review and editing

## Author ORCIDs

Ying Liu https://orcid.org/0000-0002-3328-026X

## Decision letter and Author response

Decision letter https://doi.org/10.7554/eLife.31268.026
Author response https://doi.org/10.7554/eLife.31268.027

# Additional files

## Supplementary files

• Supplementary file 1. Strains and RNAi used in this study.
DOI: https://doi.org/10.7554/eLife.31268.021

• Supplementary file 2. Lifespan anaylsis of metformin's effect on *daf-15* heterzogous mutants.
DOI: https://doi.org/10.7554/eLife.31268.022

• Supplementary file 3. Lifespan analysis of metformin's effect on mutants of TORC1 downstream genes.
DOI: https://doi.org/10.7554/eLife.31268.023

• Supplementary file 4. Lifespan analysis of metformin's effect v-ATPase, Ragulator, AXIN/LKB1 mutants.
DOI: https://doi.org/10.7554/eLife.31268.024

• Transparent reporting form
DOI: https://doi.org/10.7554/eLife.31268.025

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
