## [Decision Letter]

[Editors’ note: a previous version of this study was rejected after peer review, but the authors submitted for reconsideration. The first decision letter after peer review is shown below.]

Thank you for submitting your work entitled "Metformin Extends Lifespan by Coordination of mTOR and AMPK through Lysosomal Pathway" for consideration by *eLife*. Your article has been reviewed by three peer reviewers, one of whom is a member of our Board of Reviewing Editors and the evaluation has been overseen by Senior Editor. The reviewers have opted to remain anonymous.

Our decision has been reached after consultation between the reviewers. Based on these discussions and the individual reviews below, we regret to inform you that your work will not be considered further for publication in *eLife*.

The reviewers agreed that this is an interesting study that aims to show that metformin acts on TOR and AMPK at the lysosome, and that this site is critical for the effects of metformin on *C. elegans* lifespan. However, some of the conclusions seem to be based on small changes observed in single experiments, and analysis of more replicates with assessment of statistical significance would make the soundness of the conclusions more convincing. A substantial amount of work needs be conducted to address the concerns raised by the reviewers, and it is unlikely that this can be done within the two–month limit set by *eLife*, so we are rejecting the paper. Nevertheless, if your results hold up, I do encourage you to resubmit your work after you have obtained sufficient lines of validation.

Reviewer #1:

In this manuscript, the authors used an in vitro system to show that metformin coordinates mTORC1 inhibition and AMPK activation through the lysosomal pathway. They further demonstrated that both TORC1 inhibition and Axin/LKB1–mediated AMPK activation contribute to the lifespan extension effect of metformin in *C. elegans*. This study provides molecular mechanism of metformin–mediated lifespan extension.

Major concerns:

1) The authors showed that metformin treatment triggers nuclear localization of HLH–30::GFP in the tail region. How the cells in the tail affect the lifespan? Does loss of function of *hlh–30* suppress the lifespan extension effect of metformin?

2) Metformin treatment further extends lifespan of *daf–15* heterozygous mutants. This could be due to that metformin acts on a pathway additive to that of TORC1 inhibition. Alternatively, this may be simply due to a partial reduction in mTOR activity in *daf–15* heterozygous mutants.

3) Quantification results should be provided for Figure 1, Figure 2 and Figure 3.

Reviewer #2:

Overall, this is an interesting study that aims to show that metformin acts on TOR and AMPK at the lysosome, and that this site is critical for the effects of metformin on *C. elegans* lifespan. While a novel in vitro reconstitution assay has been developed, and several new *C. elegans* lysosomal mutants have been analyzed, the study fails to make a convincing case, mainly due to insufficient lines of validation and major rigor issues.

Lysosomal mutants:

In addition to major rigor issues (see separate box), another major concern of this article is that the authors are using several new lysosomal mutants, i.e. *lmtr–2/3* and *axl–1*, but these mutants are inadequately described and characterized. A description of the alleles, whether the mutations are loss– or gain–of function mutations, and the lysosomal function of these genes have to be established. Similar concerns relate to *vha–3* and *vha–12*, which were analyzed by RNAi. Of particular interest is whether these genes indeed localize and function within the lysosome. Lysosomal localization assays and functional assays (cathepsin assays) need to be carried out to validate these mutants, considering the rationale of the manuscript overall. Metformin's effects on lysosomal function should also be tested. Of particular concern is Figure 3, in which the authors describe that 80% of the *lmtr–2/3* mutants are censored due to vulval rupture. This censoring will greatly affect the sample size and thus the interpretation of the lifespan experiments. Again, it is crucial for the authors to share these details of their lifespan analyses for all mutants tested. The authors should also include *par–4*, the LKB1 homolog in their analyses of the lysosomal mutants.

TOR activity assays:

Throughout the manuscript, the authors are using several readouts to assay TOR activity in *C. elegans*: HLH–30 nuclear localization, DAF–16 nuclear localization, increase in *hlh–30* transcriptional reporter, increase in *gcs–1* transcriptional reporter, mRNA levels of *skn–1, daf–16* and *daf–15*/Raptor, mRNA levels of TOR pathway components, *skn–1*–dependent target genes. The authors seem to randomly choose the assay in their different experimental conditions or genetic manipulations. While we acknowledge assay limitations for TOR activity in *C. elegans*, the authors should be consistent and use the same assays throughout. To this point, analysis of S6K phosphorylation would be a direct way to assay TOR pathway status. Finally, analysis of the mRNA levels of HLH–30–regulated genes seems relevant to include, considering the use of the HLH–30 translocation assay.

TOR inhibition:

The authors use a *daf–15* heterozygous mutant to demonstrate that TOR inhibition and Metformin are additive in their effects on lifespan extension. Because the mutant is heterozygous (i.e., not null), this is not a sound conclusion, as Metformin could still exert its effects on the residual *daf–15*/TOR function. To strengthen their conclusion, the authors should rather perform epistasis experiments with known downstream targets of TOR, e.g., with *pha–4, skn–1* and *hlh–30* mutants. Additionally, the authors should assay the battery of TOR activity assay also with the *daf–15* heterozygous mutants (especially HLH–30 nuclear localization).

Fitness assays:

It would be important to include the *daf–15* heteroygous mutants in the analysis of fitness or healthspan parameters as a positive control, since metformin should still have a positive and beneficial effect on these mutants. The oil–red–o stainings only inform on decreased fat levels, however not on a mechanism by which this is achieved. The authors need to refrain from making the conclusion that Metformin activates lipolysis, as there is no data supporting this claim. Again, more information on sample size and number of repeats is needed.

in vitro reconstitution assay:

The authors establish a new in vitro reconstitution assay for lysosomal function. While this assay is innovative, it should be better validated. To this end, phosphorylation of endogenous AMPK should be assayed in all conditions. The presence of the ragulator complex (and the GTPase status if possible) should be tested as well, as provides binding for AXIN and LKB1.

The authors should discuss the possibility that AMPK directly phosphorylates Raptor and can thus directly inhibit mTORC1 (Gwinn DM et al., 2008). In light of this, it would be interesting to test whether lysosomes isolated from AMPK knock out cells would alter mTORC1 (dis)association from the lysosome.

Furthermore, the authors need to quantify their immunoblots (and add the average of their three repeats as supplemental data), since especially the Myc–Raptor binding seems to be variable in their IP:FLAG experiments.

Panel D should also include the negative control (– aas, – ConA, – Met).

Effects of metformin independent of mitochondria.

The author's conclusion that metformin acts mainly through the lysosomal pathway and not via mitochondrial inhibition is not well supported. It has been previously shown by De Haes et al., 2014, that metformin, unlike rotenone, does not increase mitochondrial stress (i.e., data included in Figure 1—figure supplement 1); in addition, they showed that metformin can inhibit electron flow from complex I, and that metformin increases ROS production (in contrast to rotenone, which decreases ROS production). The authors should therefore refrain from concluding mitochondrial–independent functions of metformin.

Reviewer #3:

While this manuscript is potentially interesting, some of the conclusions seem to be based on small changes observed in single experiments, and analysis of more replicates with assessment of statistical significance would make the soundness of the conclusions more convincing.

1) Figure 1/1E: while these results are potentially interesting, some of the observed effects are quite modest in extent, and I believe that it is essential in such cases to run at least 2 or 3 replicate incubations side–by–side so that the reproducibility of the effects can be assessed. Even better is to quantify the blots and perform statistical analysis.

2) Figure 1 was puzzled about the exact conditions in which the experiment shown in this Figure was performed. They claim to observe an increase in Thr172 phosphorylation, which would require ATP, yet it is not clear from the Methods section whether the "fractionation buffer" did contain ATP during the incubation with AXIN and LKB1. Also, did the purified Myc–LKB1 contain bound STRADalpha/β and/or MO25alpha/β, with the former in particular being essential for LKB1 activity?

3) Although this may not be common practice within the *C. elegans* community, it is possible to perform statistical analysis of survival curves such as those shown in Figure 2, Figure 3. I would like to ask why this has not been done.

[Editors’ note: what now follows is the decision letter after the authors resubmitted for further consideration.]

Thank you for submitting your article "Metformin Extends Lifespan by Coordination of mTORC1 and AMPK through Lysosomal Pathway" for consideration by *eLife*. Your article has been reviewed by two peer reviewers, one of whom is a member of our Board of Reviewing Editors and the evaluation has been overseen by Jonathan Cooper as the Senior Editor. The reviewers have opted to remain anonymous.

The reviewers have discussed the reviews with one another and the Reviewing Editor has drafted this decision to help you prepare a revised submission.

Summary:

In this manuscript, the authors showed that metformin extends lifespan through TORC1 inhibition and lysosome–dependent activation of AMPK. They developed an in vitro reconstitution assay to show that metformin coordinates mTORC1 inhibition and AMPK activation through the lysosomal pathway. They further demonstrated that both TORC1 inhibition and Axin/LKB1–mediated AMPK activation contribute to the lifespan extension effect of metformin in *C. elegans*. This study provides insights into the molecular mechanism of metformin–mediated lifespan extension. Overall, the findings are potentially interesting. This manuscript is suitable for publication in *eLife* after appropriate revisions.

Essential revisions:

1) It is unclear whether metformin–activated AMPK also could have effects on HLH–30/TFEB. The authors should directly test whether Metformin treatment of aak–2 mutants leads to HLH–30 nuclear localization (and by extension, gene expression of HLH–30 target genes). This information would help the authors in their discussion of the involvement of AMPK vs mTOR.

2) The analysis of downstream pathways of TOR is insufficiently characterized and in places contradicting with previous work from Driscoll lab. The authors performed epistasis experiments with known downstream targets of mTOR, e.g., with *pha–4, skn–1* and *hlh–30* mutants and conclude that metformin can still induce lifespan–extending effects in these mutants. However, this clearly looks like a partial lifespan extension; in the authors' hands, metformin generally produces 40% lifespan extension in wild–type/WT animals (Table S8), but in mutants they only see 14% (*hlh–30*) and 20–30% (*pha–4, smg–1*) lifespan extension (Table S6). While these lifespan experiments do not appear to have been performed in direct comparison with a WT control, the more reasonable conclusion is that metformin has a partial effect on lifespan in these backgrounds, and therefore that the genes play a partial role in the lifespan extension. Indeed, Onken and Driscoll reported that *skn–1* is required for metformin–induced longevity. The authors need to revisit their conclusions in light of these comments and previously published data. To this end, they are encouraged to emphasize the lysosome as the site of Metformin action, their most well–supported new finding, and de–focus on dissecting the individual contribution of AMPK and mTOR. A specific suggestion would to move Figure 4 to the supplements and Figure 4 merged into Figure 3.

3) While the authors have built VHA–3 and LMTR–3 reporters to confirm localization to the lysosome (were multiple independent strains analyzed and found to behave similarly?), the authors should use these strains for rescue experiments to seek possible functional validation as well.

4) The authors use a mammalian phosphorylation–specific antibody to detect phosphorylation of the worm S6K ortholog RSKS^–1^ as a read–out for TOR activity. The antibody needs to be validated.

---

## [Author Response]

[Editors’ note: the author responses to the first round of peer review follow.]

The reviewers agreed that this is an interesting study that aims to show that metformin acts on TOR and AMPK at the lysosome, and that this site is critical for the effects of metformin on C. elegans lifespan. However, some of the conclusions seem to be based on small changes observed in single experiments, and analysis of more replicates with assessment of statistical significance would make the soundness of the conclusions more convincing. A substantial amount of work needs be conducted to address the concerns raised by the reviewers, and it is unlikely that this can be done within the two–month limit set by eLife, so we are rejecting the paper. Nevertheless, if your results hold up, I do encourage you to resubmit your work after you have obtained sufficient lines of validation.

We thank the reviewers for their positive and encouraging remarks on our work. We are particularly grateful for their constructive comments that help us to enhance the clarity of the manuscript.

In the revised manuscript, we added substantial new results from six sets of experiments that address nearly every critique that the reviewers raised, as detailed below.

1) To follow the reviewers’ advice, we added several sentences in the revised manuscript to discuss that metformin treatment further extends lifespan of *daf–15* heterozygous mutants could be due to a partial reduction in mTOR activity, or due to a pathway additive to mTOR inhibition. We performed lifespan analysis with mutants of known TOR downstream targets upon metformin treatment, and also tested HLH–30 nuclear localization of *daf–15* heterozygous mutants in the presence or absence of metformin treatment. These new results suggest that metformin might act on a pathway additive to TORC1 inhibition to promote longevity.

2) In response to the suggestion that quantifications and information on experimental repeats and sample numbers should be provided, we have now provided quantifications for all immunoblots and *C. elegans* images in the revised manuscript. Information on N (population size) and n (number of repeats) are provided in figure legends. Information and statistical analysis of all lifespan experiments are provided in Supplementary files 3–8.

3) To follow the reviewer’s advice, we have tested lysosomal localization and function of all lysosomal mutants used in this manuscript, and performed additional lifespan analysis of *par–4*, the LKB1 homolog. A strain list and the verification of new mutant alleles are also provided in Supplementary file 1–Supplementary file 2.

4) In response to the reviewer’s criticism that we are using several readouts to assay TOR activity in *C. elegans*, we make it consistent and use S6K phosphorylation, HLH–30 nuclear localization, and mRNA levels of HLH–30 regulated genes to assay TOR activity in the revised manuscript.

5) We performed the analysis of fitness or healthspan parameters with *daf–15* heteroygous mutants.

6) We performed in vitro reconstitution assay in AMPK knockout cells to show that mTOR disassociation from lysosome does not require AMPK.

Reviewer #1:In this manuscript, the authors used an in vitro system to show that metformin coordinates mTORC1 inhibition and AMPK activation through the lysosomal pathway. They further demonstrated that both TORC1 inhibition and Axin/LKB1–mediated AMPK activation contribute to the lifespan extension effect of metformin in C. elegans. This study provides molecular mechanism of metformin–mediated lifespan extension.Major concerns:1) The authors showed that metformin treatment triggers nuclear localization of HLH–30::GFP in the tail region. How the cells in the tail affect the lifespan? Does loss of function of hlh–30 suppress the lifespan extension effect of metformin?

Metformin could still extend lifespan of *hlh–30* mutants, suggesting that *hlh–30* is only partially required for metformin–mediated lifespan extension (Figure 4). Neurons and intestine are two types of tissues essential for promoting longevity in *C. elegans*. We think that it is possible that the nuclear localization of HLH–30 in the tail region of the intestine is sufficient to induce lifespan extension. It is also possible that HLH–30 expresses in the nucleus of other regions or other tissues. But the fluorescent level is a bit weak for us to detect at the current exposure time.

Loss of function of *hlh–30* only partially suppresses the lifespan extension effect of metformin (Figure 4). Metformin could still extend lifespan of *hlh–30* mutants through AMPK pathway.

2) Metformin treatment further extends lifespan of daf–15 heterozygous mutants. This could be due to that metformin acts on a pathway additive to that of TORC1 inhibition. Alternatively, this may be simply due to a partial reduction in mTOR activity in daf–15 heterozygous mutants.

The reviewer is right. We added several sentences in the revised manuscript to discuss that metformin treatment further extends lifespan of *daf–15* heterozygous mutants could be due to a partial reduction in mTOR activity, or due to a pathway additive to mTOR inhibition. We followed the advice of reviewer #2 to perform lifespan experiments with mutants of known downstream targets of TOR and showed that metformin could still extend lifespans of *hlh–30, pha–4* or *skn–1* mutants (Figure 4). We also assayed HLH–30 nuclear localization with the *daf–15* heterozygous mutants and showed that *daf–15* heterozygous mutants have significant HLH–30 nuclear localization compared with control animals, and that metformin treatment does not further increase the level of HLH–30 nuclear localization in *daf–15* heterozygous mutants (Figure 3). These results suggest that metformin might act on a pathway additive to TORC1 inhibition to promote longevity.

3) Quantification results should be provided for Figure 1, Figure 2 and Figure 3.

Quantification results have now been provided in Figure 1, Figure 2 and Figure 5.

Reviewer #2:
*Overall, this is an interesting study that aims to show that metformin acts on TOR and AMPK at the lysosome, and that this site is critical for the effects of metformin on C. elegans lifespan. While a novel* in vitro reconstitution assay has been developed, and several new C. elegans lysosomal mutants have been analyzed, the study fails to make a convincing case, mainly due to insufficient lines of validation and major rigor issues.Lysosomal mutants:In addition to major rigor issues (see separate box), another major concern of this article is that the authors are using several new lysosomal mutants, i.e. lmtr–2/3 and axl–1, but these mutants are inadequately described and characterized. A description of the alleles, whether the mutations are loss– or gain–of function mutations, and the lysosomal function of these genes have to be established. Similar concerns relate to vha–3 and vha–12, which were analyzed by RNAi. Of particular interest is whether these genes indeed localize and function within the lysosome. Lysosomal localization assays and functional assays (cathepsin assays) need to be carried out to validate these mutants, considering the rationale of the manuscript overall. Metformin's effects on lysosomal function should also be tested.

We thank the reviewer for the constructive comments. Information of alleles used in this study is now provided in Supplementary file 1. We also provided the validation of *vha–3/12* and *lmtr–2/3* loss–of–function mutants in Supplementary file 2. In addition, we followed the reviewer’s advice to detect lysosomal localization and performed cathepsin assays to test lysosomal function of these mutants. These results are provided in Figure 5—figure supplement 1 and 2. Metformin’s effect on lysosomal function is also tested with cathepsin assay and provided in Figure 1—figure supplement 2.

Of particular concern is Figure 3, in which the authors describe that 80% of the lmtr–2/3 mutants are censored due to vulval rupture. This censoring will greatly affect the sample size and thus the interpretation of the lifespan experiments. Again, it is crucial for the authors to share these details of their lifespan analyses for all mutants tested.

Due to high percentage of censored worms, we enlarged sample size of *lmtr–2/3* mutants while performing the lifespan experiments. The detailed information of lifespan analysis for all mutants tested is now provided in Supplementary file 3–8.

The authors should also include par–4, the LKB1 homolog in their analyses of the lysosomal mutants.

Similar to *axl–1* (the AXIN homolog), metformin is not able to extend lifespan of *par–4* mutants as well. This data is provided in Figure 5.

TOR activity assays:Throughout the manuscript, the authors are using several readouts to assay TOR activity in C. elegans: HLH–30 nuclear localization, DAF–16 nuclear localization, increase in hlh–30 transcriptional reporter, increase in gcs–1 transcriptional reporter, mRNA levels of skn–1, daf–16 and daf–15/Raptor, mRNA levels of TOR pathway components, skn–1–dependent target genes. The authors seem to randomly choose the assay in their different experimental conditions or genetic manipulations. While we acknowledge assay limitations for TOR activity in C. elegans, the authors should be consistent and use the same assays throughout. To this point, analysis of S6K phosphorylation would be a direct way to assay TOR pathway status

We thank the reviewer for the constructive comments. In the revised manuscript, we make it consistent and use S6K phosphorylation, HLH–30 nuclear localization, and mRNA levels of HLH–30 regulated genes to assay TOR activity. These data are provided in Figure 2 and Figure 5.

Finally, analysis of the mRNA levels of HLH–30–regulated genes seems relevant to include, considering the use of the HLH–30 translocation assay.

Analysis of the mRNA levels of HLH–30–regulated genes has now been provided in Figure 2 and Figure 5.

TOR inhibition:The authors use a daf–15 heterozygous mutant to demonstrate that TOR inhibition and Metformin are additive in their effects on lifespan extension. Because the mutant is heterozygous (i.e., not null), this is not a sound conclusion, as Metformin could still exert its effects on the residual daf–15/TOR function. To strengthen their conclusion, the authors should rather perform epistasis experiments with known downstream targets of TOR, e.g., with pha–4, skn–1 and hlh–30 mutants. Additionally, the authors should assay the battery of TOR activity assay also with the daf–15 heterozygous mutants (especially HLH–30 nuclear localization).

The reviewer is right. We added several sentences in the revised manuscript to discuss that metformin treatment further extends lifespan of *daf–15* heterozygous mutants could be due to a partial reduction in TOR activity, or due to a pathway additive to TOR inhibition. We followed the reviewer’s advice to perform lifespan experiments with mutants of known downstream targets of TOR and showed that metformin could still extend lifespans of *hlh–30, pha–4* or *skn–1* loss–of–function mutants (Figure 4). We also assayed HLH–30 nuclear localization with the *daf–15* heterozygous mutants and showed that *daf–15* heterozygous mutants have significant HLH–30 nuclear localization compared with control animals, and that metformin treatment does not further increase the level of HLH–30 nuclear localization in *daf–15* heterozygous mutants (Figure 3). These results suggest that metformin might act on a pathway additive to TORC1 inhibition to promote longevity.

Fitness assays:It would be important to include the daf–15 heteroygous mutants in the analysis of fitness or healthspan parameters as a positive control, since metformin should still have a positive and beneficial effect on these mutants. The oil–red–o stainings only inform on decreased fat levels, however not on a mechanism by which this is achieved. The authors need to refrain from making the conclusion that Metformin activates lipolysis, as there is no data supporting this claim. Again, more information on sample size and number of repeats is needed.

We have performed the analysis of fitness or healthspan parameters with *daf–15* heterozygous mutants. Indeed, metformin still have a beneficial effect on these mutants. The results are provided in Figure 3.

We edited the text according to the reviewer’s suggestion to only claim that oil–red–o staining indicates decreased fat levels.

Information on sample size and number of repeats is provided in each figure legend.

in vitro reconstitution assay:The authors establish a new in vitro reconstitution assay for lysosomal function. While this assay is innovative, it should be better validated. To this end, phosphorylation of endogenous AMPK should be assayed in all conditions. The presence of the ragulator complex (and the GTPase status if possible) should be tested as well, as provides binding for AXIN and LKB1.The authors should discuss the possibility that AMPK directly phosphorylates Raptor and can thus directly inhibit mTORC1 (Gwinn DM et al., 2008). In light of this, it would be interesting to test whether lysosomes isolated from AMPK knock out cells would alter mTORC1 (dis)association from the lysosome.Furthermore, the authors need to quantify their immunoblots (and add the average of their three repeats as supplemental data), since especially the Myc–Raptor binding seems to be variable in their IP:FLAG experiments.Panel D should also include the negative control (– aas, – ConA, – Met).

Phosphorylation of endogenous AMPK and the presence of the ragulator complex have now been assayed in all conditions in the reconstitution assay. Follow the reviewer’s advice, we discussed the possibility that AMPK directly phosphorylates Raptor and can thus directly inhibit mTORC1 in the revised manuscript. We also requested AMPKα1/2 double knockout cells and repeated the in vitro assays. Lysosomes isolated from AMPK knockout cells do not alter mTORC1 (dis)association from the lysosome (Figure 1—figure supplement 1). Therefore, it is unlikely that AMPK directly inhibit mTORC1.

Quantification of the immunoblots are provided in Figure 1, and Figure 1—figure supplement 1.

We included the negative control (– aas, – ConA, – Met) in panel D.

Effects of metformin independent of mitochondria.The author's conclusion that metformin acts mainly through the lysosomal pathway and not via mitochondrial inhibition is not well supported. It has been previously shown by De Haes et al., 2014, that metformin, unlike rotenone, does not increase mitochondrial stress (i.e., data included in Figure 1—figure supplement 1); in addition, they showed that metformin can inhibit electron flow from complex I, and that metformin increases ROS production (in contrast to rotenone, which decreases ROS production). The authors should therefore refrain from concluding mitochondrial–independent functions of metformin.

We followed the reviewer’s suggestion and edited the text accordingly.

Reviewer #3:While this manuscript is potentially interesting, some of the conclusions seem to be based on small changes observed in single experiments, and analysis of more replicates with assessment of statistical significance would make the soundness of the conclusions more convincing.

We thank the reviewers for the encouraging remarks on our work and constructive comments. The statistical issue has also been raised by other reviewers. In the revised manuscript, we have now provided quantifications for all immunoblots and *C. elegans* images. Information on N (population size) and n (number of repeats) are also provided in figure legends. In addition, information and statistical analysis of survival curves are provided in Supplementary files 3–8.

1) Figure 1/1E: while these results are potentially interesting, some of the observed effects are quite modest in extent, and I believe that it is essential in such cases to run at least 2 or 3 replicate incubations side–by–side so that the reproducibility of the effects can be assessed. Even better is to quantify the blots and perform statistical analysis.

We thank the reviewer for the constructive comments. The in vitro reconstitution experiments have been repeated three times. Quantification and statistical analysis of the blots are now provided in the revised manuscript (Figure 1, and 1I).

2) Figure 1 was puzzled about the exact conditions in which the experiment shown in this Figure was performed. They claim to observe an increase in Thr172 phosphorylation, which would require ATP, yet it is not clear from the Methods section whether the "fractionation buffer" did contain ATP during the incubation with AXIN and LKB1. Also, did the purified Myc–LKB1 contain bound STRADalpha/β and/or MO25alpha/β, with the former in particular being essential for LKB1 activity?

The reviewer is correct. The “fractionation buffer” contains 2.5mM ATP. We provided this information in the revised manuscript.

Purified Myc–LKB1 does form a complex with STRAD and MO25. This result is provided in Figure 1.

3) Although this may not be common practice within the C. elegans community, it is possible to perform statistical analysis of survival curves such as those shown in Figure 2, Figure 3. I would like to ask why this has not been done.

Statistical analysis of survival curves is now provided in Supplementary files 4, 6 and 8.

[Editors' note: further revisions were requested prior to acceptance, as described below.]

Essential revisions:1) It is unclear whether metformin–activated AMPK also could have effects on HLH–30/TFEB. The authors should directly test whether Metformin treatment of aak–2 mutants leads to HLH–30 nuclear localization (and by extension, gene expression of HLH–30 target genes). This information would help the authors in their discussion of the involvement of AMPK vs mTOR.

We have showed that metformin treatment of aak–2 mutants was still able to drive HLH–30 nuclear localization and elevate the expression of HLH–30 target genes (Figure 2—figure supplement 2).

2) The analysis of downstream pathways of TOR is insufficiently characterized and in places contradicting with previous work from Driscoll lab. The authors performed epistasis experiments with known downstream targets of mTOR, e.g., with pha–4, skn–1 and hlh–30 mutants and conclude that metformin can still induce lifespan–extending effects in these mutants. However, this clearly looks like a partial lifespan extension; in the authors' hands, metformin generally produces 40% lifespan extension in wild–type/WT animals (Table S8), but in mutants they only see 14% (hlh–30) and 20–30% (pha–4, smg–1) lifespan extension (Table S6). While these lifespan experiments do not appear to have been performed in direct comparison with a WT control, the more reasonable conclusion is that metformin has a partial effect on lifespan in these backgrounds, and therefore that the genes play a partial role in the lifespan extension. Indeed, Onken and Driscoll reported that skn–1 is required for metformin–induced longevity. The authors need to revisit their conclusions in light of these comments and previously published data. To this end, they are encouraged to emphasize the lysosome as the site of Metformin action, their most well–supported new finding, and de–focus on dissecting the individual contribution of AMPK and mTOR. A specific suggestion would to move Figure 4 to the supplements and Figure 4 merged into Figure 3.

We think metformin promotes lifespan extension through two mechanisms: inhibition of TORC1 pathway and activation of AMPK. The partial lifespan extension effect of metformin in mutants of TOR downstream genes (e.g. *hlh–30, pha–4* and *skn–1*) supported our conclusions: 1) Metformin only partially extends lifespan of these mutants indicates that these TORC1 downstream genes play a partial role in the lifespan extension. It supports our idea that metformin promotes longevity partially through TOCR1 inhibition. 2) Metformin can still extends lifespans of these mutants suggests that metformin also act on a pathway additive to that of TORC1 inhibition (e.g. AMPK pathway) to confer lifespan extension.

We thank the reviewer for the suggestion to let us emphasize the lysosome as the site of metformin action, and de–focus on dissecting the individual contribution of AMPK and mTOR. We have now changed the title of our manuscript to “Metformin Extends *C. elegans* Lifespan through Lysosomal Pathway”. We also follow the reviewer’s specific suggestion to move Figure 4 to Figure 3—figure supplement 2 and Figure 4 to Figure 3.

3) While the authors have built VHA–3 and LMTR–3 reporters to confirm localization to the lysosome (were multiple independent strains analyzed and found to behave similarly?), the authors should use these strains for rescue experiments to seek possible functional validation as well.

Yes, the VHA–3 and LMTR–3 localization were analyzed with multiple independent transgenic strains. They all behave similarly.

We have followed the reviewer’s suggestion to rescue *vha–3, vha–12, lmtr–2* and *lmtr–3* mutants with the corresponding reporter construct and validate their function through cathepsin assays (Figure 4—figure supplement 2). The reporter construct can rescue the mutant phenotype.

4) The authors use a mammalian phosphorylation–specific antibody to detect phosphorylation of the worm S6K ortholog RSKS^–1^ as a read–out for TOR activity. The antibody needs to be validated.

We have now validated the antibody with rsks^–1^ RNAi (Figure 2—figure supplement 1).